# Preparation and Characterization of Bio-oil Phenolic Foam Reinforced with Montmorillonite

**DOI:** 10.3390/polym11091471

**Published:** 2019-09-09

**Authors:** Pingping Xu, Yuxiang Yu, Miaomiao Chang, Jianmin Chang

**Affiliations:** 1College of Materials Science and Technology, Beijing Forestry University, Beijing 100083, China (P.X.) (M.C.); 2College of Art and Design, Zhejiang Sci-Tech University, Hangzhou 310018, China

**Keywords:** phenolic foam, bio-oil, montmorillonite, toughness, flame resistance

## Abstract

Introducing bio-oil into phenolic foam (PF) can effectively improve the toughness of PF, but its flame retardant performance will be adversely affected and show a decrease. To offset the decrease in flame retardant performance, montmorillonite (MMT) can be added as a promising alternative to enhance the flame resistance of foams. The present work reported the effects of MMT on the chemical structure, morphological property, mechanical performance, flame resistance, and thermal stability of bio-oil phenolic foam (BPF). The Fourier transform infrared spectroscopy (FT-IR) result showed that the –OH group peaks shifted to a lower frequency after adding MMT, indicating strong hydrogen bonding between MMT and bio-oil phenolic resin (BPR) molecular chains. Additionally, when a small content of MMT (2–4 wt %) was added in the foamed composites, the microcellular structures of bio-oil phenolic foam modified by MMT (MBPFs) were more uniform and compact than that of BPF. As a result, the best performance of MBPF was obtained with the addition of 4 wt % MMT, where compressive strength and limited oxygen index (LOI) increased by 31.0% and 33.2%, respectively, and the pulverization ratio decreased by 40.6% in comparison to BPF. These tests proved that MMT can blend well with bio-oil to effectively improve the flame resistance of PF while enhancing toughness.

## 1. Introduction

Polymeric foams, such as polyurethane (PUR), polyisocyanurate (PIR), polyvinyl chloride (PVC), and phenolic foams (PFs), have been widely researched and applied in a variety of emerging sectors [1,2,3,4,5]. Meanwhile, recent development of reinforced polymeric foams, has gained increasing interest driven by increasingly stringent requirements of high-performance foams in current markets. Besides, the advancement of nanotechnology has become an important driving force in the development of nanocomposite foams, endowing the polymeric foams with more functional characteristics [4,6]. Research on reinforced foams has focused on improving thermal and mechanical properties, alongside other properties [3,4,5].

PFs, known as the “king of insulation materials”, have also received considerable attention, in this regard of reinforced foams, thanks to their low thermal conductivity, good thermal insulation, excellent flame resistance, low smoke density and nontoxicity, and no dripping in combustion [7]. However, the application of traditional PFs are severely restricted by their high brittleness and pulverization, thus many efforts have been made to strengthen them through the incorporation of reinforcements [8,9,10,11,12]. The consistently increasing energy demand and environmental awareness justify the development of rising efforts to seek new renewable and effective toughening agents for PFs [13]. Numerous researchers have utilized natural raw materials such as lignin [14,15,16], tannins [17,18], cardanol [19,20], and bio-oil [21] to toughen PFs. However, the excellent flame retardant performance of PF is negatively affected by the addition of these toughening agents. Among them, bio-oil, an eco-friendly renewable liquid product obtained by fast pyrolysis of biomass, is rich in phenols, ketones, aldehydes, ethers, acids, and hydrocarbons [22]. Some phenolic compounds in bio-oil with a flexible chained structure, such as guaiacol, can react with formaldehyde leading to the increased toughness of PFs [21,23,24,25]. Additionally, partial substitution of phenol with bio-oil can also ease the dependence on non-renewable petroleum resources and further reduce the cost. In our previous work, we successfully prepared phenolic foams modified with bio-oil (BPFs), proving that bio-oil could be a renewable toughener for PFs, but reduced the flame resistance of PFs, like most toughening agents [21]. Thus, the research on maintaining flame resistance while also enhancing toughness is important for BPFs.

Nano-montmorillonite (MMT), another natural recourse, is extensively used to obtain increased flame resistance and thermal stability of polymer composites due to its thermal barrier properties and retarding effect on thermal degradation [26,27]. Meanwhile, significant performance improvements can be achieved at lower MMT loading, which can be more environmentally friendly and competitive. Another impressive feature of MMT is the concurrent improvement of multiple properties, such as mechanical property [28]. Thus, MMT is a good material to enhance the flame resistance of phenolic resin and its foams [29,30]. Several works have demonstrated the suitability of MMT to improve the flame resistance of polymer foams [31,32]. However, little research has been carried out on the modification of BPF by MMT.

The aim of this paper is to improve the toughness of PF using bio-oil while ensuring its excellent flame resistance with the addition of MMT. Therefore, in this study, the phenolic resins modified with bio-oil and MMT (MBPRs) were prepared and their basic performances were tested. Furthermore, the BPF and phenolic foams modified with bio-oil and MMT (MBPFs) were prepared and their mechanical performance, morphological property, flame resistance, and thermal stability were assessed by mechanical testing, scanning electron microscopy (SEM), limited oxygen index (LOI) analysis, and thermogravimetric analysis (TGA), respectively. Meanwhile the structure of MBPFs was investigated by Fourier transform infrared spectroscopy (FT-IR) to identify the interactions between MMT and phenolic resin modified with bio-oil (BPR).

## 2. Materials and Methods 

### 2.1. Materials

Bio-oil, obtained in a fluidized bed at 550 °C via the fast pyrolysis of Larix gmelinii (Rupr.) Kuzen, was made at the Lab of Fast Pyrolysis of Biomass and Productive Utilization (Beijing Forestry University, Beijing, China). Bio-oil is a complex mixture which contains water (30 wt %) and a broad range of organic compounds, including 33.42% phenols, 29.56% ketones, 12.45% aldehydes, 10.05% polysaccharides, 9.33% organic acids, and 4.39% other compounds, such as esters and ethers. Phenol, paraformaldehyde, p-toluenesulfonic acid monohydrate and montmorillonite K-10 were purchased from Shanghai Macklin Biochemical Co., Ltd., Shanghai, China. Sodium hydroxide (NaOH), petroleum ether (30–60), and phosphoric acid were provided by Beijing Chemical Industries Reagent Co., Ltd., Beijing, China. Tween-80 was supplied by Guanghua Chemical Reagent Co., Ltd., Guangdong, China.

### 2.2. Synthesis and Characterization of MBPRs

The foaming MBPRs with different MMT (wt % to phenol) addition rates (2, 4, 6, and 8 wt %) were synthesized using a molar ratio of phenol (including bio-oil)/paraformaldehyde/NaOH of 1:1.8:0.4 and a bio-oil/P substitution rate of 30%. Firstly, phenol and 75 wt % NaOH solution (concentration of 30%) were placed in a 500 mL three-necked flask. Then, the 75 wt % paraformaldehyde was slowly added, and the temperature of the mixture was heated to 65–70 °C. Following the addition of 25 wt % NaOH solution, the mixture was kept at 65–70 °C for 15 min, then heated to 90 °C and held for 20 min. Second, the pre-prepared mixture of bio-oil and MMT (after 30 min of ultrasonication) and the residual 25 wt % paraformaldehyde were added, and the temperature of the system was maintained at 80 °C for 20 min. Third, the temperature was rapidly cooled down to 40 °C in 20 min, and the hydrochloric acid (37 wt %) was used to adjust the pH value of the mixture to between 6.0 and 7.2 to yield the resins. These prepared resins were denoted as 2% MBPR, 4% MBPR, 6% MBPR, and 8% MBPR. In addition, the pristine BPR (the control resin) without MMT was prepared with the same method and coded as BPR. 

The viscosity of resin was determined at 25 °C using NDJ-5S rotating viscometer (CANY, Shanghai, China). The solid content and curing time of resins were measured based on China National Standards (GB/T 14074-2017) [33]. Wherein, the curing time of MBPR was determined in a foaming process at 75 °C in the same composite curing agents (p-toluenesulfonic acid/phosphoric acid) so as to simulate the curing environment and obtain the accurate curing time. In this test, 7.5 ± 0.01 g of curing agent was added into the 50 ± 0.1 g of resin and the mixture was thoroughly stirred under ambient temperature. Then 10 ± 0.1 g of the mixture was rapidly placed in a test tube and kept stirring in the water bath at 75 °C, and the time was recorded when the stir bar could not move. Each of the above tests was repeated at least three times. The basic characterizations of BPR and MBPRs are listed in Table 1.

### 2.3. Preparation of MBPFs

The surfactant and blowing agent was Tween-80 and petroleum ether, respectively. The composite curing agents used a molar ratio of p-toluenesulfonic acid/phosphoric acid of 2:1. Based on the mass of resin, 5 wt % surfactants, 8 wt % blowing agents, and 15 wt % composite curing agents were sequentially added into the foamable BPR and MBPRs. Then the mixture was loaded in a mold and rapidly mixed well for 3 min by stirring at room temperature. Subsequently, the stirred mixture was rapidly placed into a preheated oven and bubbled at 75 °C for 40 min. These prepared foams were denoted as BPF, 2% MBPF, 4% MBPF, 6% MBPF, and 8% MBPF.

### 2.4. Characterization

FT-IR analysis of the MMT, BPF, and MBPFs were performed directly with a Nicolet iS5 FT-IR (Thermo Fisher Scientific, Waltham, MA, USA) to investigate the variation of functional groups within the range of 400–4000 cm^−1^.

SEM imaging of the cross-sectional morphologies of the foam samples was carried out with a microanalyzer of a SU8010 SEM (Hitachi, Tokyo, Japan) to observe the microstructure of the foam. The cell sizes of the different samples were measured from SEM micrographs using the Nanomeasure 1.2 software (Keithley Instruments, Inc., Cleveland, OH, USA) to systematically analyze the variations.

The apparent density of foam was measured based on the China National Standard (GB/T 6343-2009) [34] with sample size of 50 × 50 × 50 mm^3^ (width × length × thickness). The average of five samples was calculated. 

The pulverization ratio was carried out in accordance with the China National Standard (GB/T 12812-2006) [35]. In this test, the sample (30 × 30 × 30 mm^3^) pressed with 200 g weight was pushed back and forth 30 times on a 300-mesh abrasive paper with a constant force and the distance of per one-way friction was 250 mm. Meanwhile, the specimens were weighted before and after testing, and the weight loss of a sample after friction was applied to measure the pulverization ratio as follows [36]: (1)$$\mathrm{Pulverization}\;\mathrm{ratio}(\%)=\frac{m_{0}-m_{1}}{m_{0}}\times 100\%$$
where *m*_0_ is the original mass and m_1_ is the final mass of the foam. 

The compressive property of the foams was tested on a universal testing machine (Instron, Havisham, England) in accordance to the China National Standard (GB/T 8813-2008) [37]. The samples were cut into the dimensions of 50 × 50 × 50 mm^3^. Compressive strength was obtained as the maximum strength of the stress taken at 10% deformation with a crosshead rate of 2.5 mm/min via the test machine computer control software. At least five replicates were used for this test. Then the data of the compressive strength was normalized for density (the compressive strength was divided by the density of the samples) to eliminate the differing densities between samples.

The LOI test of the foams was determined by using a YZS-8A full automatic oxygen index tester (Xinshengzhuoyue Analysis Instrument Co., Beijing, China) according to China National Standard (GB/T 2406-2009) [38]. A minimum of five replicates with a size of 10 × 10 × 100 mm^3^ of each sample were tested. 

TG analysis of the foams was used to investigate the thermal stability and examined in N_2_ flow heated from 0 to 800 °C at a heating rate of 10 °C/min using a Q5000IR analyzer (TA Instruments, New Castle, DE, USA). 

## 3. Results and Discussion

### 3.1. FT-IR Analysis

The FT-IR spectra of MMT, the BPF and MBPFs are illustrated in Figure 1, and the functional groups corresponding to the major peaks are identified and listed in Table 2 [21,23,39,40,41]. With respect to BPF, the characteristic absorption peaks at 3315 and 1612 cm^−1^ are ascribed to phenol O–H stretching and aromatic skeletal vibration, respectively. The absorption peaks at 2917 and 2866 cm^−1^ are attributed to CH_2_ stretching vibration. For MMT, the peak at 3625 cm^−1^ is attributed to the –OH stretching mode bending of Al–OH, while the signal at 3418 cm^−1^ corresponds to Si–OH bending of MMT. Another large band observed around 1035 cm^−1^ is assigned to the Si–O stretching (in-plane) vibration of the silicate framework. As seen in Figure 1, all of the modified MBPFs present similar curves in comparison to BPF, indicating that these foams have similar chemical structures. Interestingly, some differences between the MBPFs and BPF are also found. In the MBPFs′ spectra, the absorption peak detected at 3625 cm^−1^ became more obvious with the increase of the MMT addition rate, possibly due to more Al–OH in the MBPFs after the incorporation of MMT. Moreover, after the addition of MMT, the O–H stretching vibration showed broader peaks and a red shift from 3315 cm^−1^ to 3307 and 3279 cm^−1^ compared with BPF. This was likely due to the formation of hydrogen bonding between the OH groups in BPR chains and the layer surface O atoms (Si–O of MMT). The possible schematic of hydrogen bond formation between BPR and MMT is demonstrated in Figure 2. Similarly, this hydrogen bonding between the surface of MMT platelets and OH groups was also found in earlier studies [42,43]. Meanwhile, it should be noted that in the 4% MBPF spectra, the shift of the peak from 3315 (O–H stretching vibration) to 3279 cm^−1^ was more obvious compared with other spectra, indicating stronger hydrogen bonding between the BPR chains and MMT. In addition, as reported in the literature [44,45], the strong interaction between the BPR chains and MMT corresponded to the good dispersion of MMT in the materials. In other word, the well dispersed MMT with a larger surface area of dispersed phase is beneficial to form hydrogen bonding [45].

### 3.2. Foam Morphology of MBPFs

As shown in Figure 3, the SEM measurement was performed to analyze the foam morphology and cell size distribution. The mean cell size of BPF and MBPFs with different MMT addition rates are summarized in Table 3. It can be seen in Figure 3 that closed cells were mostly demonstrated for all samples. Compared with BPF, the cell size distributions were narrower and a more compact and uniform microcellular structure was observed in 2% MBPF and 4% MBPF. This phenomenon is attributed to the fact that MMT, a nucleating agent, could effectively promote the heterogeneous cell nucleation. Meanwhile, since MMT provided higher melt strength and spreading speed in the foam, it formed homogeneous cells [46]. In contrast, in the case of 6% MBPF and 8% MBPF, the cell size distributions were wider and the cell structures were non-uniform as compared to the BPF, possibly because of the MMT agglomeration resulting from its high surface area and energy [47]. As shown in Table 3, the mean cell size of 2–8% MBPF becomes significantly smaller as the amount of MMT increases. The plausible explanations for this result are that (i) the increased viscosity (Table 1) due to the addition of MMT, and (ii) the diffusion barrier because of the high aspect ratio of MMT both impose restrictions on the expansion of the cells [32,48]. Notably, after incorporating MMT into BPR, the MPBFs exhibited a larger bubble size compared to BPF. This phenomenon can be attributed to the longer curing time (Table 1) of MBPFs in relation to that of BPF, which meant that the resins did not reach the appropriate viscosity in time, thus the films between bubbles were too weak to prevent merging and agglomeration. As seen in Table 1, the viscosity and curing time of MBPRs increased with the increase of MMT compared to BPR because of the layer structure of MMT, which is an obstacle in the formation and movement of network structures.

### 3.3. Apparent Density, Pulverization Rate, and Compressive Strength

Figure 4 shows the apparent density, pulverization rate, and compressive strength of the BPF and MBPFs. As seen, the apparent density of MBPFs first decreased and then rose with the increase of MMT addition rate, showing a minimum at 4 wt % MMT addition rate. Similar results of the reduction of foam density at lower loadings of MMT were also reported in other literature [49,50,51], and this trend can be explained by the opposing effect of MMT addition on nucleation and cell growth, which was also reported in Madaleno’s research [47]. A similar trend is shown between pulverization rate and apparent density. The pulverization rate of 4% MBPF reached its lowest value and decreased by 40.6% from 8.76% to 5.20% in relation to that of BPF. This was primarily related to the high MMT compatibility with the BPR resin [52], meanwhile, a strong hydrogen bonding interaction existed between the BPR chains and MMT in 4% MBPF according to the FT-IR result. However, with the MMT addition rate increasing to 6 wt %, the MMT exhibited low compatibility with BPR owing to its agglomeration, leading to a high pulverization rate. The compressive strength of MBPFs first increased and then decreased with the increase of MMT addition rate. Foam with a 4 wt % addition of MMT showed an increase of 31% in the compressive strength in comparison with that of BPF. The visible improvement in compressive strength may be attributed in part to the good interaction between the particles and the matrix [53], and the smaller and more uniform cells of 4% MBPF than other foams [54]. Meanwhile, good dispersion of MMT in 4% MBPF can achieve more efficient load transfer, which lead to the improved compressive strength [32]. Nevertheless, with the MMT addition rate increasing to 6 wt %, the compressive strength of MBPF was negatively affected by the aggregation of MMT and non-uniform cells of MBPF (Figure 3).

### 3.4. Limited Oxygen Index Analysis

LOI is an important parameter which is widely used to assess the flammability of materials. The effect of MMT on the fire resistance of BPF was investigated by LOI testing. As shown in Figure 5, the BPF presented the lowest LOI value at 29.8%, indicating poor inflammability. Fortunately, the addition of MMT into the BPF system induced significant improvement on the flame resistance for MBPFs, and all the LOI values were higher than the B1 standard value (≥30%), according to the China National Standards GB/T 8624-2012 [55] and the European Standard EN13501-1:2007 [56], which means that MBPFs have superior flame-retardancy. The LOI values initially increased with the increasing addition rate of MMT until a maximum was attained and then decreased. The MBPF exhibited the highest LOI value at 39.7% when MMT was loaded at 4 wt %, leading to a 33.2% improvement compared to that of BPF. Several factors could explain the strengthening inflammability effect of the MMT modifier to BPF. First, the tortuous paths created by the complex orientation of MMT layers reduced heat radiation and conduction [57]. Second, the interaction between MMT and BPR increased the viscosity, inhibiting the droplet in the combustion process and limiting the spread of the flame. Third, the further formed MMT-based char layer at high temperature had a more compact and orderly intercalation structure, thereby acting as an effective protective shield on the BPF surface to delay the transfer of oxygen, heat, and volatile gases between the flame and underlying matrix. Last, the well dispersed MMT improved the compatibility between MMT and BPR [58].

### 3.5. Thermal Stability

The thermal stability of MBPFs was assessed by the TG and derivative thermogravimetric analysis (DTG) in an atmosphere of nitrogen. As shown in Figure 6, thermograms obtained were similar in the studied BPF and MBPFs, and the thermal degradation of BPF and MBPFs could be roughly divided into three stages. The first degradation stage occurred below 100 °C, which was attributed to the evaporation of residual moisture, blowing agents, and volatiles such as phenol, or formaldehyde in the foams [59]. The second step corresponded to the decomposition and evaporation of the surfactant (Tween-80) and curing agents, as well as the release of small molecules owing to the further curing of BPR and MBPR in the range of 100–350 °C. Meanwhile, in this stage, the C–O–C bridges were broken and converted to CH_2_ bridges due to further curing. The third stage occurring from 350 to 800 °C was the main degradation caused by the chain scission such as the decomposition of bridged methylene and the further degradation of phenols to carbonaceous structures. 

The effect of MMT on the thermal degradation of BPF can be followed. TG curves of MBPFs shifted to high temperature, which implied that the addition of MMT can improve the thermal stability of BPF. In order to further understand the obtained results for the materials, the thermogram for MMT was also studied. Below 150 °C, the MMT exhibited a degradation stage attributed to the removal of adsorbed and interlayer water [60], and then it remained stable in the range of 150–550 °C, and then began to degrade above 550 °C. Therefore, when MMT was incorporated in the BPF, MBPFs degraded faster than BPF in the range of 0–100 °C, but the temperatures of fastest decomposition of MPBFs in this stage were still increased owing to the excellent thermal stability of MMT. Furthermore, the noticeable difference between BPF and MBPFs was in the range of temperature of 100–350 °C, where the C–O–C bridges of BPF degraded significantly faster than that of MBPF. Notably, the fastest decomposition temperatures of MBPFs in this second stage were remarkably increased by about 40 °C in relation to that of BPF. These phenomena indicated that MMT played an important role on the second degradation step, directly slowing down the escape of degraded compounds and the diffusion of oxygen [61]. 

From Table 4, *T*_5%_ (at which 5 wt % mass loss had taken place) and *T*_max_ of the MBPFs were all higher than those of BPF. The MBPF exhibited the highest *T*_5%_ and *T*_max_ value at 179 and 501 °C, respectively, when MMT was loaded at 4 wt %. Additionally, a slight increase was found in residual mass at 800 °C with MMT incorporation. The improved thermal stability of MBPFs can be attributed to the following two reasons: (i) The activity of the BPR macromolecule chains was clearly confined by the stronger hydrogen bond interaction between BPR molecules and MMT, thus it required more energy for segmental motion [62]; and (ii) MMT layers, as superior insulators and mass transport barriers, can effectively suppress the conduction of heat and the emanation of the small molecule when a degradation reaction originated, decreasing the degradation velocity of the composites and limiting further degradation [63,64]. In conclusion, there was an optimal MMT addition rate at 4 wt % in this system for the best thermal stability of MBPFs, which was in accordance with the LOI test.

## 4. Conclusions

MBPFs showed remarkably enhanced toughness as well as good flame resistance compared to BPF. Introducing MMT can form the hydrogen bonding between the O–H groups of BPR and Si–O groups of MMT without altering the chemical structure, which can enhance the interaction between the particles and the matrix on the basis of FTIR analysis. Meanwhile, MMT was also found to increase the cell size and simultaneously form a more uniform structure as shown by SEM micrographs. Thereby, these can be proven by the lower pulverization ratio and higher compressive strength of 2% MBPF and 4% MBPF, in comparison with BPF. In conjunction, LOI and TG results further proved the improvement of flame resistance and thermal stability of MBPFs. To conclude, the preparation of MBPF described in this study could provide a new and natural modifier for the BPF and provide an efficient way to reduce fire hazards and ensure improved toughness.

## Figures and Tables

**Figure 1 polymers-11-01471-f001:**
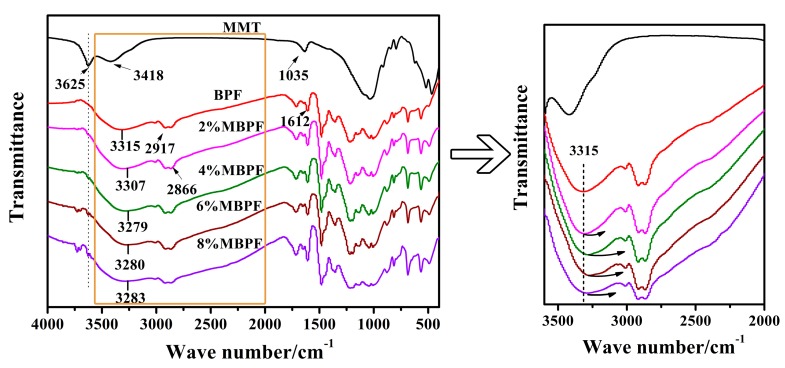
FT-IR of montmorillonite (MMT), BPF and MBPFs.

**Figure 2 polymers-11-01471-f002:**
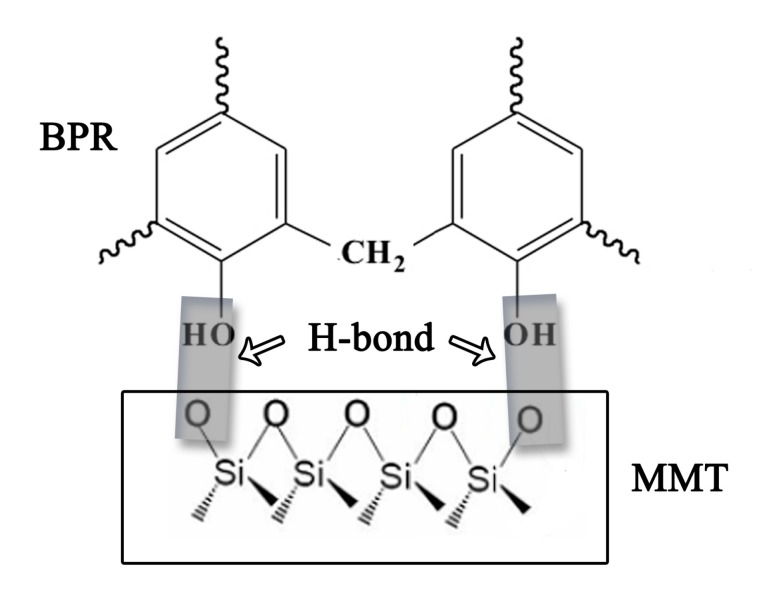
Schematic of hydrogen bond formation between BPR and MMT.

**Figure 3 polymers-11-01471-f003:**
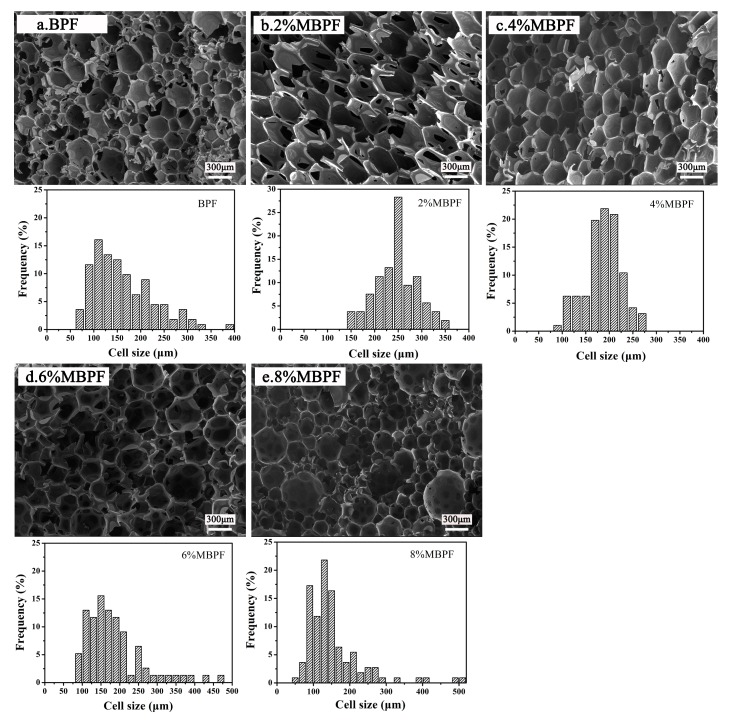
SEM images and cell size distributions of BPF and MBPFs: (**a**) BPF; (**b**) 2% MBPF; (**c**) 4% MBPF; (**d**) 6% MBPF; (**e**) 8% MBPF.

**Figure 4 polymers-11-01471-f004:**
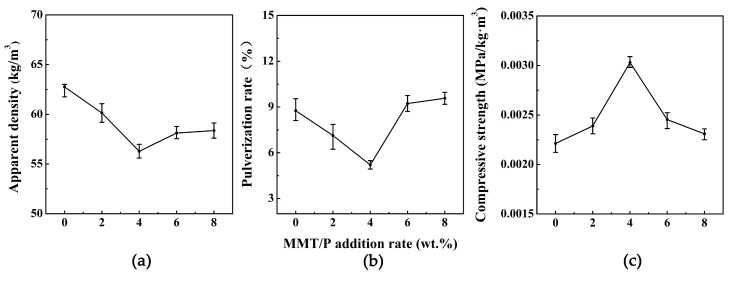
(**a**) Apparent density; (**b**) pulverization rate and (**c**) compressive strength of BPF and MBPFs.

**Figure 5 polymers-11-01471-f005:**
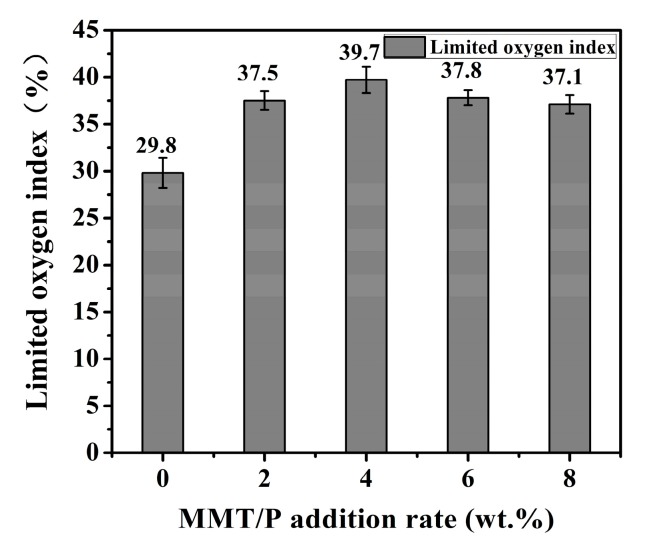
LOI analysis of BPF and MBPFs.

**Figure 6 polymers-11-01471-f006:**
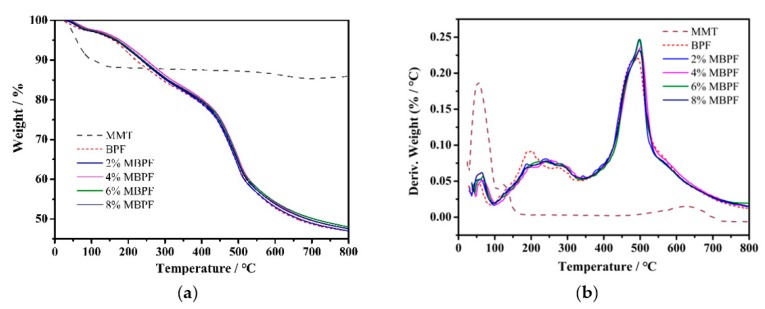
(**a**) TG and (**b**) DTG curves of BPF, MBPFs, and MMT.

**Table 1 polymers-11-01471-t001:** Basic characteristics of bio-oil phenolic resin (BPR) and bio-oil phenolic resins modified by montmorillonite (MBPRs).

Samples	Viscosity (25 °C, mPa·s)	Solids Content (%)	Curing Time (75 °C, s)
BPR	2840 ± 38	72.0 ± 0.6	254 ± 10
2% MBPR	2930 ± 60	73.5 ± 0.3	624 ± 12
4% MBPR	3050 ± 87	76.3 ± 0.1	811 ± 36
6% MBPR	7250 ± 56	74.9 ± 0.6	817 ± 32
8% MBPR	10800 ± 75	73.5 ± 0.3	818 ± 20

**Table 2 polymers-11-01471-t002:** Peaks and assignment of FT-IR spectra for MMT, BPF and MBPFs.

Wave Number (cm^−1^)	Vibration	Assignment
3625	ν (OH)	Al–OH stretching vibration
3418	ν (OH)	Si–OH stretching vibration
3315	ν (OH)	Phenolic OH and aliphatic OH stretching vibration
2917,2866	ν (CH_2_)	Aliphatic CH_2_ asymmetric stretching vibration
1612	ν (C=C)	C=C aromatic ring stretching vibration
1035	ν (Si–O)	Si–O stretching (in-plane) vibration

ν: Stretching vibration.

**Table 3 polymers-11-01471-t003:** Mean cell size of BPF and MBPFs.

Samples	BPF	2% MBPF	4% MBPF	6% MBPF	8% MBPF
Mean Cell size (μm)	162 ± 36	246 ± 22	188 ± 10	185 ± 31	174 ± 14

**Table 4 polymers-11-01471-t004:** The onset temperature of decomposition (*T*_5%_), the temperature of fastest decomposition (*T*_max_), and the residual mass at 800 °C for BPF and MBPFs.

Samples	*T*_5%_ (°C)	*T*_max_ (°C)	Residue at 800 °C (%)
BPF	160	492	46.99
2% MBPF	166	498	47.03
4% MBPF	179	501	47.50
6% MBPF	171	499	47.93
8% MBPF	165	498	47.55

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
