# Peer review of "Preparation and Characterization of Bio-oil Phenolic Foam Reinforced with Montmorillonite"

_polymers, 2019, doi:10.3390/polym11091471_

Round 1

Reviewer 1 Report

General comments
Badly and casually drafted manuscript. Reject and Resubmit.

Specific comments
Abstract: In lines 13 to 17 authors state that "The effects of MMT on chemical structure, morphological properties, mechanical performance, flame resistance and thermal stability of BPF were studied by Fourier transform infrared spectrometry (FT-IR), scanning electron microscopy (SEM), apparent density, pulverization rate, compressive strength, limited oxygen index (LOI) and thermogravimetric (TG) analysis." How can SEM give information about LOI or FTIR about morphology. Rephrase this ridiculous statement.

Did authors read their manuscript before submitting. The lines 29 to 36 "The introduction should briefly place the study in a broad context and highlight why it is.. " appears to be that of template. Authors are supposed to remove these lines before submitting.

In section 3.4. Limited oxygen index analysis authors state that "according to the GB/T 8624-2012 of the China National Standards" This journal is an international journal. Authors are strongly recommended to use more acceptable international standards like ASTM or DIN.

However, a google search about "GB/T 8624-2012" gave the Chinese standard available from the website: www.chinesestandard.net/PDF/English.aspx/GB8624-2012. 
In lines 236 and 237 authors also state that "LOI values of them were higher than the B1 standard value (≥ 32%)", but Table 7 of GB 8624-2012 titled "Table 7 Grades and Classification Criteria of Burning Behavior of Foamed Plastics for Electrical Apparatus and Furniture Product" doesn't mention about LOI, but only states "Heat release rate peak value of unit area ≤400 kW/m2; Mean burning time ≤30 s, Mean burning height ≤250 mm" How did author's correlate their LOI values to mean burning time and mean burning height. Explain its methodology in the revised manuscript.

There are many more problems in the manuscript. But these are enough to recommend reject & resubmit.

Reviewer 2 Report

In this work, bio-oil phenolic foams modified by montmorillonite were successfully prepared. Furthermore, the effects of montmorillonite on chemical structure, morphological properties, mechanical performance, flame resistance and thermal stability of bio-oil phenolic foam were investigated. The overall balance and structure of the paper is good, however at a closer reading the following improvements are recommended:
-The Abstract section it needs to be written more concisely.
-The first paragraph of the Introduction section must be deleted (lines 29-36). I think the authors wrote it by mistake.
-In this paper the authors presents a preparation/characterization of reinforced phenolic foams. In the opinion of the reviewer, a brief overview (one or two paragraphs) of other types of reinforced foams (PUR, PIR, PVC, etc.) would increase the value of the work. There are certain research groups (Antunes et al., Linul et al., Andersons et al., Marsavina et al. etc.) dealing with the physical and mechanical characterization of different reinforced foam materials. Please refer to their works in the Introduction section.
-I think formula (1) can be deleted because it is well known by all researchers in the field and beyond.
-Each chapter or sub-chapter must begin with text, not with a figure.
-The used standards must be presented also in the References list.
-A figure with samples (before and after tests) is encouraged to be presented.
-The standard deviations should be added to Table 3.
-Some tabular results are encouraged to be presented.
-Some recent developments published in the Polymers journal should be considered, showing a continuity between the present work and those reported in the literature on similar topics (e.g doi.org/10.3390/polym11081267, doi.org/10.3390/polym11071192, doi.org/10.3390/polym10121298, doi.org/10.3390/polym11061028).
-English is not the native language of this Reviewer; however, the manuscript requires a strong proof reading.
